# Uniform Convergence of Interpolators: Gaussian Width, Norm Bounds and Benign Overfitting

**Frederic Koehler**[*]
MIT
fkoehler@mit.edu

**Lijia Zhou**[*]
University of Chicago
zlj@uchicago.edu

**Danica J. Sutherland**
UBC and Amii
dsuth@cs.ubc.ca

**Nathan Srebro**
TTI-Chicago
nati@ttic.edu

Collaboration on the Theoretical Foundations of Deep Learning (deepfoundations.ai)

## Abstract

We consider interpolation learning in high-dimensional linear regression with Gaussian data, and prove a generic uniform convergence guarantee on the generalization error of interpolators in an arbitrary hypothesis class in terms of the class's Gaussian width. Applying the generic bound to Euclidean norm balls recovers the consistency result of Bartlett et al. (2020) for minimum-norm interpolators, and confirms a prediction of Zhou et al. (2020) for near-minimal-norm interpolators in the special case of Gaussian data. We demonstrate the generality of the bound by applying it to the simplex, obtaining a novel consistency result for minimum $\ell_1$-norm interpolators (basis pursuit). Our results show how norm-based generalization bounds can explain and be used to analyze benign overfitting, at least in some settings.

## 1 Introduction

The traditional understanding of machine learning suggests that models with zero training error tend to overfit, and explicit regularization is often necessary to achieve good generalization. Given the empirical success of deep learning models with zero training error (Neyshabur et al. 2015; Zhang et al. 2017) and the (re-)discovery of the "double descent" phenomenon (Belkin et al. 2019), however, it has become clear that the textbook U-shaped learning curve is only part of a larger picture: it is possible for an overparameterized model with zero training loss to achieve low population error in a noisy setting. In an effort to understand how interpolation learning occurs, there has been much recent study of the testbed problem of linear regression with Gaussian features (e.g. Hastie et al. 2019; Bartlett et al. 2020; Belkin et al. 2020; Ju et al. 2020; Muthukumar et al. 2020; Negrea et al. 2020; Tsigler and Bartlett 2020; Zhou et al. 2020). Significant progress has been made in this setting, including nearly-matching necessary and sufficient conditions for consistency of the minimal $\ell_2$ norm interpolator (Bartlett et al. 2020).

Despite the fundamental role of uniform convergence in statistical learning theory, most of this line of work has used other techniques to analyze the particular minimal-norm interpolator.[1] Instead of directly analyzing the population error of a learning algorithm, a uniform convergence-type argument would control the worst-case generalization gap over a class of predictors containing the typical outputs of a learning rule. Typically, this is done because for many algorithms – unlike the minimal Euclidean norm interpolator – it is difficult to exactly characterize the learned predictor, but we may be able to say e.g. that its norm is not too large. Since uniform convergence does not tightly depend on a specific algorithm, the resulting analysis can highlight the key properties that lead to good generalization: it can give bounds not only for, say, the minimal-norm interpolator, but also

---

[*]These authors contributed equally.

[1]Negrea et al. (2020) argue that Bartlett et al. (2020)'s proof technique is fundamentally based on uniform convergence of a *surrogate* predictor; Yang et al. (2021) study a closely related setting with a uniform convergence-type argument, but do not establish consistency. We discuss both papers in more detail in Section 4.

35th Conference on Neural Information Processing Systems (NeurIPS 2021).

for other interpolators with low norm (e.g. Zhou et al. 2020), increasing our confidence that low norm – and not some other property the particular minimal-norm interpolator happens to have – is key to generalization. In linear regression, practical training algorithms may not always find the exact minimal Euclidean norm solution, so it is also reassuring that all interpolators with sufficiently low Euclidean norm generalize.

Nagarajan and Kolter (2019), however, raised significant questions about the applicability of typical uniform convergence arguments to certain high-dimensional regimes, similar to those seen in interpolation learning. Following their work, Bartlett and Long (2020), Negrea et al. (2020), Zhou et al. (2020), and Yang et al. (2021) all demonstrated the failure of forms of uniform convergence in various interpolation learning setups. To sidestep these negative results, Zhou et al. (2020) suggested considering bounds which are uniform only over predictors *with zero training error*. This weaker notion of uniform convergence has been standard in analyses of "realizable" (noiseless) learning at least since the work of Vapnik (1982, Chapter 6.4) and Valiant (1984). Zhou et al. demonstrated that at least in one particular noisy setting, such uniform convergence is sufficient for showing consistency of the minimal $\ell_2$ norm interpolator, even though "non-realizable" uniform convergence arguments (those over predictors regardless of their training error) cannot succeed. It remains unknown, however, whether these types of arguments can apply to more general linear regression problems and more typical asymptotic regimes, particularly showing rates of convergence rather than just consistency.

In this work, we show for the first time that uniform convergence is indeed able to explain benign overfitting in general high-dimensional Gaussian linear regression problems. Similarly to how the standard analysis for learning with Lipschitz losses bounds generalization gaps through Rademacher complexity (e.g. Shalev-Shwartz and Ben-David 2014), our Theorem 1 (Section 3) establishes a finite-sample high probability bound on the uniform convergence of the error of interpolating predictors in a hypothesis class, in terms of its Gaussian width. This is done through an application of the Gaussian Minimax Theorem; see the proof sketch in Section 7. Combined with an analysis of the norm of the minimal $\ell_2$ norm interpolator (Theorem 2 in Section 4), our bound recovers known consistency results (Bartlett et al. 2020), as well as proving a conjectured upper bound for larger-norm interpolators (Zhou et al. 2020).

In addition, since we do not restrict ourselves to Euclidean norm balls but instead consider interpolators in an arbitrary compact set, our results allows for a wide range of other applications. Our analysis leads to a natural extension of the consistency result and notions of effective rank of Bartlett et al. (2020) for arbitrary norms (Theorem 5 in Section 5). As a demonstration of our general theory, in Section 6 we show novel consistency results for the minimal $\ell_1$ norm interpolator (basis pursuit) in particular settings, which we believe are the first results of their kind.

## 2   Problem Formulation

**Notation.**   We use $\|\cdot\|_p$ for the $\ell_p$ norm, $\|x\|_p = (\sum_i |x_i|^p)^{1/p}$. We always use $\max_{x \in S} f(x)$ to be $-\infty$ when $S$ is empty, and similarly $\min_{x \in S} f(x)$ to be $\infty$. We use standard $O(\cdot)$ notation, and $a \lesssim b$ for inequality up to an absolute constant. For a positive semidefinite matrix $A$, the *Mahalanobis (semi-)norm* is $\|x\|_A^2 := \langle x, Ax \rangle$. For a matrix $A$ and set $S$, $AS$ denotes the set $\{Ax : x \in S\}$.

**Data model.**   We assume that data $(X, Y)$ is generated as

$$Y = Xw^* + \xi, \qquad X_i \overset{iid}{\sim} N(0, \Sigma), \qquad \xi \sim N(0, \sigma^2 I_n), \tag{1}$$

where $X \in \mathbb{R}^{n \times d}$ has i.i.d. Gaussian rows $X_1, \ldots, X_n$, $d \geq n$, $w^*$ is arbitrary, and $\xi$ is Gaussian and independent of $X$. Though our proof techniques crucially depend on $X_i$ being Gaussian, we can easily relax the assumption on the noise $\xi$ to only being sub-Gaussian; we assume Gaussian noise here for simplicity. The *empirical* and *population loss* are defined as, respectively,

$$\hat{L}(w) = \frac{1}{n} \|Y - Xw\|_2^2, \quad L(w) = \mathop{\mathbb{E}}_{(x,y)} (y - \langle w, x \rangle)^2 = \sigma^2 + \|w - w^*\|_\Sigma^2,$$

where in the expectation $y = \langle x, w^* \rangle + \xi_0$ with $x \sim N(0, \Sigma)$ independent of $\xi_0 \sim N(0, \sigma^2)$. For an arbitrary norm $\|\cdot\|$, the minimal norm interpolator is $\hat{w} = \arg\min_{\hat{L}(w)=0} \|w\|$. For Euclidean norm specifically, the minimal norm interpolator can be written explicitly as $\hat{w} = X(XX^T)^{-1}Y$. If there is more than one minimal norm interpolator, all of our guarantees will hold for any minimizer $\hat{w}$.

**Speculative bound.** Zhou et al. (2020) studied uniform convergence of low norm interpolators,

$$\sup_{\|w\| \leq B,\, \hat{L}(w)=0} L(w) - \hat{L}(w). \tag{2}$$

Clearly, when $B \geq \|\hat{w}\|$, this quantity upper-bounds the population risk of $\hat{w}$. Zhou et al. evaluated the asymptotic limit of (2) in one particular setting. But they further speculated that a bound of the following form may hold more generally:

$$\sup_{\|w\|_2 \leq B,\, \hat{L}(w)=0} L(w) - \hat{L}(w) \leq \frac{B^2 \psi_n}{n} + o(1), \tag{$\star$}$$

where[2] $\psi_n \approx \|x\|^2$ . As discussed by Zhou et al., a bound almost of this form is implied by results of Srebro et al. (2010) for general data distributions, except that approach gives a large leading constant and logarithmic factors. To show consistency of benign overfitting, though, we need ($\star$) to hold without even a constant multiplicative factor. Zhou et al. ask whether and when this holds, speculating that it might in broad generality.

The goal of this paper is essentially to prove ($\star$), at least for Gaussian data, and to use it to show consistency of the minimal norm interpolator. Our main result (Theorem 1) can be thought of showing ($\star$) for Gaussian data with $\psi_n = \mathbb{E} \|x\|^2$, as well as strengthening and significantly generalizing it. Our result is more general, as it applies to general compact hypothesis sets beyond just the Euclidean ball as in ($\star$). But it also falls short of fully proving the speculative ($\star$) since our results are limited to Gaussian data, while there is no reason we are aware of to believe a tight uniform convergence guarantee of the form ($\star$) does not hold much more broadly. We leave extending our results beyond the Gaussian case open.

## 3  Generic Uniform Convergence Guarantee

To state our results, we first need to introduce some key tools.

**Definition 1.** The *Gaussian width* and the *radius* of a set $S \subset \mathbb{R}^d$ are

$$W(S) := \mathop{\mathbb{E}}_{H \sim N(0, I_d)} \sup_{s \in S} |\langle s, H \rangle| \quad \text{and} \quad \mathrm{rad}(S) := \sup_{s \in S} \|s\|_2.$$

The radius measures the size of a set in the Euclidean norm. The Gaussian width of a set $S$ can be interpreted as the number of dimensions that a random projection needs to approximately preserve the norms of points in $S$ (Gordon 1988; Bandeira 2016). These two complexity measures are connected by Gaussian concentration: Gaussian width is the expected value of the supremum of some Gaussian process, and the radius upper bounds the typical deviation of that supremum from its expected value.

**Definition 2** (Covariance splitting). Given a positive semidefinite matrix $\Sigma \in \mathbb{R}^{d \times d}$, we write $\Sigma = \Sigma_1 \oplus \Sigma_2$ if $\Sigma = \Sigma_1 + \Sigma_2$, each matrix is positive semidefinite, and their spans are orthogonal.

Note that $\Sigma = \Sigma_1 \oplus \Sigma_2$ effectively splits the eigenvectors of $\Sigma$ into two disjoint parts.

We can now state our generic bound. Section 7 sketches the proof; all full proofs are in the appendix.

**Theorem 1** (Main generalization bound). *There exists an absolute constant $C_1 \leq 66$ such that the following is true. Under the model assumptions in* (1), *let $\mathcal{K}$ be an arbitrary compact set, and take any covariance splitting $\Sigma = \Sigma_1 \oplus \Sigma_2$. Fixing $\delta \leq 1/4$, let $\beta = C_1 \left( \sqrt{\frac{\log(1/\delta)}{n}} + \sqrt{\frac{\mathrm{rank}(\Sigma_1)}{n}} \right)$. If $n$ is large enough that $\beta \leq 1$, then the following holds with probability at least $1 - \delta$:*

$$\sup_{w \in \mathcal{K},\, \hat{L}(w)=0} L(w) \leq \frac{1+\beta}{n} \left[ W(\Sigma_2^{1/2} \mathcal{K}) + \left( \mathrm{rad}(\Sigma_2^{1/2} \mathcal{K}) + \|w^*\|_{\Sigma_2} \right) \sqrt{2 \log \left( \frac{32}{\delta} \right)} \right]^2.$$

In our applications, we consider $\mathcal{K} = \{w \in \mathbb{R}^d : \|w\| \leq B\}$ for an arbitrary norm, with $B$ based on a high-probability upper bound for $\|\hat{w}\|$. Depending on the application, the rank of $\Sigma_1$ will be

---

[2]When $\|x\|^2$ concentrates, we need $\psi_n$ to match its typical value. That is, $\psi_n$ might be a high probability bound on $\|x\|^2$, or for (sub)Gaussian data, as in our case, $\psi_n = \mathbb{E} \|x\|^2$.

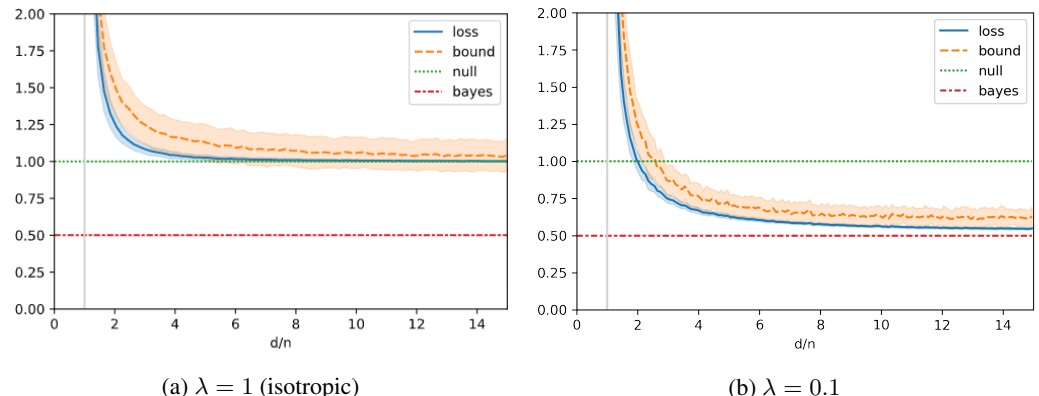

(a) $\lambda = 1$ (isotropic)       (b) $\lambda = 0.1$

Figure 1: Illustration of our generalization bound when $\Sigma = \begin{bmatrix} 1 & 0 \\ 0 & \lambda^2 I_{d-1} \end{bmatrix}$, $n = 200$, $\sigma^2 = 1/2$, $w^* = (1/\sqrt{2}, 0, \ldots, 0)$, and $d$ is varied (x-axis). Averages (curve) and standard deviations (error bars) are estimated from 400 trials for each value of $d$. Here the curve marked "loss" corresponds to $L(\hat{w})$ for the minimum Euclidean norm interpolator $\hat{w}$, "bound" to $\|\hat{w}\|_2^2 \operatorname{Tr} \Sigma / n = \mathbb{E}\|\hat{w}\|_2^2 (1 + \lambda^2(d-1))/n$ which is an asymptotic bound on $L(\hat{w})$ due to Corollary 1, "null" is the loss $L(0) = 1$ of the zero estimator, and "bayes" is the Bayes-optimal error $L(w^*) = \sigma^2 = 1/2$. The vertical line is $d/n = 1$, the location of the double-descent peak; for $d/n < 1$ there are almost surely no interpolators.

either constant or $o(n)$, so that $\beta \to 0$. The term $\|w^*\|_{\Sigma_2}$ generally does not scale with $n$ and hence is often negligible. As hinted earlier, we can think of the Gaussian width term and the radius term as bias and variance, respectively. To achieve consistency, we can expect that the Gaussian width should scale as $\sigma \sqrt{n}$. This agrees with the intuition that we need increasing norm to memorize noise when the model is not realizable. The radius term requires some care in our applications, but can be handled by the covariance splitting technique. As part of the analysis in the following sections, we will rigorously show in many settings that the dominant term in the upper bound is the Gaussian width. In these cases, our upper bound is roughly $W(\Sigma_2^{1/2}\mathcal{K})^2/n$, which can be viewed as the ratio between the (probabilistic) dimension of our hypothesis class and sample size. We will also analyze how large $\mathcal{K}$ must be to contain any interpolators, allowing us to find consistency results.

## 4  Application: Euclidean Norm Ball

It can be easily seen that the Gaussian width of a Euclidean norm ball reduces nicely to the product of the norm of our predictor with the typical norm of $x$: if $\mathcal{K} = \{w \in \mathbb{R}^d : \|w\|_2 \le B\}$, then

$$W(\Sigma^{1/2}\mathcal{K}) = B \cdot \mathop{\mathbb{E}}_{H \sim N(0, I_d)} \|\Sigma^{1/2} H\|_2 \le \sqrt{B^2 \mathbb{E}\|x\|_2^2}. \tag{3}$$

Therefore, it is plausible that ($\star$) holds with $\psi_n = \mathbb{E}\|x\|_2^2 = \operatorname{Tr}(\Sigma)$. Figure 1 illustrates this generalization bound in two simple examples, motivated by Hastie et al. (2019) and Zhou et al. (2020). Indeed, an application of our main theorem proves that this is exactly the case for Gaussian data.

**Corollary 1** (Proof of the speculative bound ($\star$) for Gaussian data). *Fix any $\delta \le 1/4$. Under the model assumptions in (1) with $B \ge \|w^*\|_2$ and $n \gtrsim \log(1/\delta)$, for some $\gamma \lesssim \sqrt[4]{\log(1/\delta)/n}$, it holds with probability at least $1 - \delta$ that*

$$\sup_{\|w\|_2 \le B, \hat{L}(w)=0} L(w) \le (1 + \gamma) \frac{B^2 \operatorname{Tr}(\Sigma)}{n}. \tag{4}$$

The above bound is clean and simple, but only proves a sub-optimal rate of $n^{-1/4}$. This is because the choice of covariance split used in the proof of Corollary 1 uses no information about the particular structure of $\Sigma$. This bound can also be slightly loose in situations where the eigenvalues of $\Sigma$ decay

rapidly, in which case $\mathrm{Tr}(\Sigma)$ can be replaced by a smaller quantity. We next state a more precise bound on the generalization error, which requires introducing the following notions of effective rank.

**Definition 3** (Bartlett et al. 2020)**.** The *effective ranks* of a covariance matrix $\Sigma$ are

$$r(\Sigma) = \frac{\mathrm{Tr}(\Sigma)}{\|\Sigma\|_{op}} \quad \text{and} \quad R(\Sigma) = \frac{\mathrm{Tr}(\Sigma)^2}{\mathrm{Tr}(\Sigma^2)}.$$

The $r(\Sigma)$ rank can roughly be understood as the squared ratio between the Gaussian width and radius in our previous bound. It is related to the concentration of the $\ell_2$ norm of a Gaussian vector with covariance $\Sigma$. In fact, both definitions of effective ranks can be derived by applying Bernstein's inequality to $\|x\|^2 / \mathbb{E}\|x\|^2$. We will only need $r(\Sigma)$ in the generalization bound below, but we will show in Theorem 2 that $R(\Sigma)$ can be used to control the norm of the minimal norm interpolator $\hat{w}$.

**Corollary 2.** *There exists an absolute constant $C_1 \le 66$ such that the following is true. Under* (1)*, pick any split $\Sigma = \Sigma_1 \oplus \Sigma_2$, fix $\delta \le 1/4$, and let $\gamma = C_1 \left( \sqrt{\frac{\log(1/\delta)}{r(\Sigma_2)}} + \sqrt{\frac{\log(1/\delta)}{n}} + \sqrt{\frac{\mathrm{rank}(\Sigma_1)}{n}} \right)$. If $B \ge \|w^*\|_2$ and $n$ is large enough that $\gamma \le 1$, the following holds with probability at least $1 - \delta$:*

$$\sup_{\|w\|_2 \le B, \hat{L}(w)=0} L(w) \le (1 + \gamma)\frac{B^2 \mathrm{Tr}(\Sigma_2)}{n}. \tag{5}$$

In order to use Corollary 1 or 2 to prove consistency, we need a high-probability bound for $\|\hat{w}\|_2$, the norm of the minimal norm interpolator, so that $B$ will be large enough to contain any interpolators. Theorem 2 gives exactly such a bound, showing that if the effective ranks $R(\Sigma_2)$ and $r(\Sigma_2)$ are large, then we can construct an interpolator with Euclidean norm nearly $\|w^*\|_2 + \sigma\sqrt{n/\mathrm{Tr}(\Sigma_2)}$.

**Theorem 2** (Euclidean norm bound; special case of Theorem 4)**.** *Fix any $\delta \le 1/4$. Under the model assumptions in* (1) *with any choice of covariance splitting $\Sigma = \Sigma_1 \oplus \Sigma_2$, there exists some $\epsilon \lesssim \sqrt{\frac{\log(1/\delta)}{r(\Sigma_2)}} + \sqrt{\frac{\log(1/\delta)}{n}} + \frac{n\log(1/\delta)}{R(\Sigma_2)}$ such that the following is true. If $n$ and the effective ranks are such that $\epsilon \le 1$ and $R(\Sigma_2) \gtrsim \log(1/\delta)^2$, then with probability at least $1 - \delta$, it holds that*

$$\|\hat{w}\|_2 \le \|w^*\|_2 + (1 + \epsilon)^{1/2}\, \sigma\sqrt{\frac{n}{\mathrm{Tr}(\Sigma_2)}}. \tag{6}$$

Plugging in estimates of $\|\hat{w}\|$ to our scale-sensitive bound Corollary 2, we obtain a population loss guarantee for $\hat{w}$ in terms of effective ranks.

**Theorem 3** (Benign overfitting)**.** *Fix any $\delta \le 1/2$. Under the model assumptions in* (1) *with any covariance splitting $\Sigma = \Sigma_1 \oplus \Sigma_2$, let $\gamma$ and $\epsilon$ be as defined in Corollary 2 and Theorem 2. Suppose that $n$ and the effective ranks are such that $R(\Sigma_2) \gtrsim \log(1/\delta)^2$ and $\gamma, \epsilon \le 1$. Then, with probability at least $1 - \delta$,*

$$L(\hat{w}) \le (1 + \gamma)(1 + \epsilon)\left( \sigma + \|w^*\|_2\sqrt{\frac{\mathrm{Tr}(\Sigma_2)}{n}} \right)^2. \tag{7}$$

From (7), we can see that to ensure consistency, i.e. $L(\hat{w}) \to \sigma^2$, it is enough that $\gamma \to 0$, $\epsilon \to 0$, and $\|w^*\|_2 \sqrt{\frac{\mathrm{Tr}(\Sigma_2)}{n}} \to 0$. Recalling the definitions of the various quantities and using that $r(\Sigma_2)^2 \ge R(\Sigma_2)$ (Bartlett et al. 2020, Lemma 5), we arrive at the following conditions.

**Sufficient conditions for consistency of $\hat{w}$.** As $n \to \infty$, $L(\hat{w})$ converges in probability to $\sigma^2$ if there exists a sequence of covariance splits $\Sigma = \Sigma_1 \oplus \Sigma_2$ such that

$$\frac{\mathrm{rank}(\Sigma_1)}{n} \to 0, \qquad \|w^*\|_2 \sqrt{\frac{\mathrm{Tr}(\Sigma_2)}{n}} \to 0, \qquad \frac{n}{R(\Sigma_2)} \to 0. \tag{8}$$

**Relationship to Bartlett et al. (2020).** Our set of sufficient conditions above subsumes and is slightly more general than the conditions of Bartlett et al. (2020). There are two differences:

1. They choose the covariance split specifically to minimize $\mathrm{rank}(\Sigma_1)$ such that $r(\Sigma_2) \gtrsim n$.

2. Their version of the second condition replaces $\mathrm{Tr}(\Sigma_2)$ by the larger term $\mathrm{Tr}(\Sigma)$.

From the perspective of showing $L(\hat{w}) \to \sigma^2$, the first difference is immaterial: if there exists a choice of split that satisfies our conditions, it can be shown that there exists a (possibly different) split which will also satisfy $r(\Sigma_2) \gtrsim n$ (see Appendix D.2.1). The second point is a genuine improvement over the consistency result of Bartlett et al. (2020) when $\Sigma$ has a few very large eigenvalues; this improvement has also been implicitly obtained by Tsigler and Bartlett (2020).

Regarding the rate of convergence, our additional $r(\Sigma_2)^{-1/2}$ term and the dependence on $\sqrt{\mathrm{rank}(\Sigma_1)/n}$ instead of $\mathrm{rank}(\Sigma_1)/n$ is slightly worse than that of Bartlett et al. (2020), but our bound can be applied for a smaller value of $\mathrm{rank}(\Sigma_1)$ and is better in the $\|w^*\|_2 \sqrt{\mathrm{Tr}(\Sigma_2)/n}$ term. We believe these differences are minimal in most cases, and not so important for our primary goal to showcase the power of uniform convergence.

**Relationship to Negrea et al. (2020).** The consistency result of Bartlett et al. (2020) can also be recovered with a uniform convergence-based argument (Negrea et al. 2020). Instead of considering uniform convergence over a norm ball, Negrea et al. applied uniform convergence to a surrogate predictor, and separately showed that the minimal-norm interpolator has risk close to the surrogate (and, indeed, argue that this was fundamentally the proof strategy of Bartlett et al. (2020) all along). Their analysis reveals an interesting connection between realizability and interpolation learning, but it does not highlight that low norm is key to good generalization, nor does it predict the worst-case error for other low-norm interpolators.

**Relationship to Bartlett and Long (2020).** Bartlett and Long recently showed that it is impossible to find a tight excess risk bound that only depends on the learned predictor and sample size. This does not, however, contradict our results. A closer look at the construction of their lower bound reveals that the excess risk bounds being ruled out cannot depend on either the training error $\hat{L}$ or the population noise level $\sigma^2$. The former is crucial: their considered class of bounds cannot incorporate the knowledge that the training error is small, which is the defining property of uniform convergence of interpolators. The latter point is also important; they consider excess risk $(L - \sigma^2)$, but ($\star$) and our bounds are about the generalization gap $(L - \hat{L})$.

**Relationship to Yang et al. (2021).** Yang et al. give expressions for the asymptotic generalization error of predictors in a norm ball, in a random feature model. Their model is not directly comparable to ours (their labels are effectively a nonlinear function of their non-Gaussian random features), but they similarly showed that uniform convergence of interpolators can lead to a non-vacuous bound. It is unclear, though, whether uniform convergence of low-norm interpolators can yield consistency in their model: they only study sets of the form $\{\|w\| \leq \alpha \|\hat{w}\|\}$ with $\alpha > 1$ a constant, where we would expect a loss of $\alpha^2 \sigma^2$ – i.e. would not expect consistency. They also rely on numerical methods to compare their (quite complicated) analytic expressions. It remains possible that the gap between uniform convergence of interpolators and the Bayes risk vanishes in their setting as $\alpha$ approaches 1.

# 5 General Norm Ball

All the results on the Euclidean setting are special cases of the following results for arbitrary norms. It is worth keeping in mind that the Euclidean norm will still play a role in these analyses, via the Gaussian width and radius appearing in Theorem 1 and the $\ell_2$ projection $P$ appearing in Theorem 4.

**Definition 4.** The *dual norm* of a norm $\|\cdot\|$ on $\mathbb{R}^d$ is $\|u\|_* := \max_{\|v\|=1} \langle v, u \rangle$, and the set of all its sub-gradients with respect to $u$ is $\partial \|u\|_* = \{v : \|v\| = 1, \langle v, u \rangle = \|u\|_*\}$.

The Euclidean norm's dual is itself; for it and many other norms, $\partial \|u\|_*$ is a singleton set. Using these notions, we will now give versions of the effective ranks appropriate for generic norm balls.

**Definition 5.** The effective $\|\cdot\|$-ranks of a covariance matrix $\Sigma$ are given as follows. Let $H \sim N(0, I_d)$, and define $v^* = \arg\min_{v \in \partial \|\Sigma^{1/2} H\|_*} \|v\|_\Sigma$. Then

$$ r_{\|\cdot\|}(\Sigma) = \left( \frac{\mathbb{E} \left\| \Sigma^{1/2} H \right\|_*}{\sup_{\|w\| \leq 1} \|w\|_\Sigma} \right)^2 \quad \text{and} \quad R_{\|\cdot\|}(\Sigma) = \left( \frac{\mathbb{E} \left\| \Sigma^{1/2} H \right\|_*}{\mathbb{E} \|v^*\|_\Sigma} \right)^2 . $$

The first effective $\|\cdot\|$-rank is the squared ratio of Gaussian width to the radius of the set $\Sigma^{1/2}\mathcal{K}$, where $\mathcal{K}$ is a norm ball $\{w : \|w\| \leq B\}$; the importance of this ratio should be clear from Theorem 1. The Gaussian width is given by $W(\Sigma^{\frac{1}{2}}\mathcal{K}) = B\,\mathbb{E}\,\|x\|_*$, while the radius can be written $\sup_{\|w\|\leq B}\|w\|_\Sigma$ so that the factors of $B$ cancel.

The choice of $R_{\|\cdot\|}$ arises naturally from our bound on $\|\hat{w}\|$ in Theorem 4 below. Large effective rank of $\Sigma_2$ means the sub-gradient $v^*$ of $\|\Sigma_2^{1/2}H\|_*$ is small in the $\|\cdot\|_{\Sigma_2}$ norm. This is, in fact, closely related to the existence of low-norm interpolators. First, note that $\Sigma_2^{1/2}H$ corresponds to the small-eigenvalue components of the covariate vector. For $v^*$ to be a sub-gradient means that moving the weight vector $w$ in the direction of $v^*$ is very effective at changing the prediction $\langle w, X\rangle$; having small $\|\cdot\|_{\Sigma_2}$ norm means that moving in this direction has a very small effect on the population loss $L(w)$. Together, this means the sub-gradient will be a good direction for benignly overfitting the noise.

**Remark 1** (Definitions of effective ranks). Using $\|\cdot\|_2$ in Definition 5 yields slightly different effective ranks than those of Definition 3, but the difference is small and asymptotically negligible. Both $r$ and $R$ use $\mathbb{E}\,\|x\|^2$ in their numerators, while $r_{\|\cdot\|_2}$ and $R_{\|\cdot\|_2}$ use $(\mathbb{E}\,\|x\|)^2$. The denominators of $r$ and $r_{\|\cdot\|_2}$ agree; Lemma 9, in Appendix C.2, shows that $r(\Sigma) - 1 \leq r_{\|\cdot\|_2}(\Sigma) \leq r(\Sigma)$. The denominator of $R_{\|\cdot\|_2}$ uses the sole sub-gradient $v^* = \Sigma^{1/2}H/\|\Sigma^{1/2}H\|_2$; the denominator is then $\|\Sigma^{1/2}H\|_\Sigma^2/\|\Sigma^{1/2}H\|_2^2 \approx \mathrm{Tr}(\Sigma^2)/\mathrm{Tr}(\Sigma)$, giving that $R_{\|\cdot\|_2}(\Sigma) \approx \mathrm{Tr}(\Sigma)^2/\mathrm{Tr}(\Sigma^2) = R(\Sigma)$. Equation (75), in Appendix C.2, shows that $R_{\|\cdot\|_2}(\Sigma) \geq cR(\Sigma)$ for some $c$ that converges to 1 as $r(\Sigma) \to \infty$, as is required by the consistency conditions. The other direction, $R(\Sigma) \geq c'R_{\|\cdot\|_2}(\Sigma)$, also holds with $c' \to 1$ as $r(\Sigma^2) \to \infty$. It would be possible (and probably even more natural with our analysis) to state the consistency conditions in Section 4 in terms of $r_{\|\cdot\|_2}$ and $R_{\|\cdot\|_2}$; we used $r$ and $R$ mainly to allow direct comparison to Bartlett et al. (2020).

Using the general notion of effective ranks, we can find an analogue of Corollary 2 for general norms.

**Corollary 3.** *There exists an absolute constant $C_1 \leq 66$ such that the following is true. Under the model assumptions in (1), take any covariance splitting $\Sigma = \Sigma_1 \oplus \Sigma_2$ and let $\|\cdot\|$ be an arbitrary norm. Fixing $\delta \leq 1/4$, let $\gamma = C_1\left(\sqrt{\frac{\log(1/\delta)}{r_{\|\cdot\|}(\Sigma_2)}} + \sqrt{\frac{\log(1/\delta)}{n}} + \sqrt{\frac{\mathrm{rank}(\Sigma_1)}{n}}\right)$. If $B \geq \|w^*\|$ and $n$ is large enough that $\gamma \leq 1$, then the following holds with probability at least $1 - \delta$:*

$$\sup_{\|w\|\leq B,\hat{L}(w)=0} L(w) \leq (1+\gamma)\frac{\left(B \cdot \mathbb{E}\|\Sigma_2^{1/2}H\|_*\right)^2}{n}. \tag{9}$$

As in the Euclidean special case, we still need a bound on the norm of $\hat{w} = \arg\min_{\hat{L}(w)=0}\|w\|$ to use this result to study the consistency of $\hat{w}$. This leads us to the second main technical result of this paper, which essentially says that if the effective ranks $R_{\|\cdot\|}(\Sigma_2)$ and $r_{\|\cdot\|}(\Sigma_2)$ are sufficiently large, then there exists an interpolator with norm $\|w^*\| + \sigma\sqrt{n}/\mathbb{E}\|\Sigma_2^{1/2}g\|_*$.

**Theorem 4** (General norm bound). *There exists an absolute constant $C_2 \leq 64$ such that the following is true. Under the model assumptions in (1) with any covariance split $\Sigma = \Sigma_1 \oplus \Sigma_2$, let $\|\cdot\|$ be an arbitrary norm, and fix $\delta \leq 1/4$. Denote the $\ell_2$ orthogonal projection matrix onto the space spanned by $\Sigma_2$ as $P$. Let $H \sim N(0, I_d)$, and let $v^* = \arg\min_{v\in\partial\|\Sigma_2^{1/2}H\|_*}\|v\|_{\Sigma_2}$. Suppose that there exist $\epsilon_1, \epsilon_2 \geq 0$ such that with probability at least $1 - \delta/4$*

$$\|v^*\|_{\Sigma_2} \leq (1+\epsilon_1)\,\mathbb{E}\,\|v^*\|_{\Sigma_2} \qquad \text{and} \qquad \|Pv^*\|^2 \leq 1 + \epsilon_2; \tag{10}$$

*let $\epsilon = C_2\left(\sqrt{\frac{\log(1/\delta)}{r_{\|\cdot\|}(\Sigma_2)}} + \sqrt{\frac{\log(1/\delta)}{n}} + (1+\epsilon_1)^2\frac{n}{R_{\|\cdot\|}(\Sigma_2)} + \epsilon_2\right)$. Then if $n$ and the effective ranks are large enough that $\epsilon \leq 1$, with probability at least $1 - \delta$, it holds that*

$$\|\hat{w}\| \leq \|w^*\| + (1+\epsilon)^{1/2}\,\sigma\,\frac{\sqrt{n}}{\mathbb{E}\,\|\Sigma_2^{1/2}H\|_*}. \tag{11}$$

For a specific choice of norm $\|\cdot\|$, we can verify that $\|v^*\|_{\Sigma_2}$ is small. In the Euclidean case, for example, this is done by (78) in Appendix C.2; in our basis pursuit application to come, this is done

by (92). The term $\epsilon_2$ measures the cost of using a projected version of the subgradient; in most of our applications, we can take $\epsilon_2 = 0$. Recalling that $\|v^*\| = 1$, this is obviously true with the Euclidean norm for any $\Sigma_2$. More generally, if $\Sigma$ is diagonal, then it is natural to only consider covariance splits $\Sigma = \Sigma_1 \oplus \Sigma_2$ such that $\Sigma_2$ is diagonal. Then, when $\|\cdot\|$ is the $\ell_1$ norm (or $\ell_p$ norms more generally), it can be easily seen that $Pv^* = v^*$ and so $\|Pv^*\| = \|v^*\| = 1$.

Straightforwardly combining Corollary 3 and Theorem 4 yields the following theorem, which gives guarantees for minimal-norm interpolators in terms of effective rank conditions. Just as in the Euclidean case, we can extract from this result a simple set of sufficient conditions for consistency of the minimal norm interpolator.

**Theorem 5** (Benign overfitting with general norm). *Fix any $\delta \leq 1/2$. Under the model assumptions in (1), let $\|\cdot\|$ be an arbitrary norm and pick a covariance split $\Sigma = \Sigma_1 \oplus \Sigma_2$. Suppose that $n$ and the effective ranks are sufficiently large such that $\gamma, \epsilon \leq 1$ with the same choice of $\gamma$ and $\epsilon$ as in Corollary 3 and Theorem 4. Then, with probability at least $1 - \delta$,*

$$L(\hat{w}) \leq (1 + \gamma)(1 + \epsilon) \left( \sigma + \|w^*\| \frac{\mathbb{E} \|\Sigma_2^{1/2} H\|_*}{\sqrt{n}} \right)^2. \tag{12}$$

**Sufficient conditions for consistency of $\hat{w}$.** As $n \to \infty$, $L(\hat{w})$ converges in probability to $\sigma^2$ if there exists a sequence of covariance splits $\Sigma = \Sigma_1 \oplus \Sigma_2$ such that

$$\frac{\text{rank}(\Sigma_1)}{n} \to 0, \qquad \frac{\|w^*\| \mathbb{E} \|\Sigma_2^{1/2} H\|_*}{\sqrt{n}} \to 0, \qquad \frac{1}{r_{\|\cdot\|}(\Sigma_2)} \to 0, \qquad \frac{n}{R_{\|\cdot\|}(\Sigma_2)} \to 0, \tag{13}$$

and, with the same definition of $P$ and $v^*$ as in Theorem 4, it holds for any $\eta > 0$ that

$$\Pr(\|Pv^*\|^2 > 1 + \eta) \to 0. \tag{14}$$

As we see, the conditions for a minimal norm interpolator to succeed with a general norm generalize those from the Euclidean setting in a natural way. As discussed above, (14) is always satisfied for the Euclidean norm. The only remaining notable difference from the Euclidean setting is that we have two large effective dimension conditions on $\Sigma_2$ instead of a single one; in the Euclidean case, the condition on $R$ implies the condition on $r$.

## 6 Application: $\ell_1$ Norm Balls for Basis Pursuit

The theory for the minimal $\ell_1$ norm interpolator, $\hat{w}_{BP} \in \arg\min_{\hat{L}(w)=0} \|w\|_1$ – also known as *basis pursuit* (Chen et al. 2001) – is much less developed than that of the minimal $\ell_2$ norm interpolator. In this section, we illustrate the consequences of our general theory for basis pursuit. Full statements and proofs of results in this section are given in Appendix E.

The dual of the $\ell_1$ norm is the $\ell_\infty$ norm $\|u\|_\infty = \max_i |u_i|$, and $\partial \|u\|_\infty$ is the convex hull of $\{\text{sign}(u_i) e_i : i \in \arg\max |u_i|\}$. From the definition of sub-gradient, we observe that

$$\min_{v \in \partial \|\Sigma^{1/2} g\|_\infty} \|v\|_\Sigma \leq \max_{i \in [d]} \|e_i\|_\Sigma = \sqrt{\max_i \Sigma_{ii}}. \tag{15}$$

Furthermore, by convexity we have

$$\max_{\|w\|_1 \leq 1} \|w\|_\Sigma = \sqrt{\max_i \langle e_i, \Sigma e_i \rangle} = \sqrt{\max_i \Sigma_{ii}} \tag{16}$$

and so $r_{\|\cdot\|_1}(\Sigma) = \frac{(\mathbb{E} \|\Sigma^{1/2} g\|_\infty)^2}{\max_i \Sigma_{ii}} \leq R_{\|\cdot\|_1}(\Sigma)$. Therefore, we can use a single notion of effective rank. For simplicity, we denote $r_1(\Sigma) = r_{\|\cdot\|_1}(\Sigma)$. Combining this with (13) and the previous discussion of (14), we obtain the following sufficient conditions for consistency of basis pursuit.

**Sufficient conditions for consistency of $\hat{w}_{BP}$.** As $n \to \infty$, $L(\hat{w})$ converges to $\sigma^2$ in probability if there exists a sequence of covariance splits $\Sigma = \Sigma_1 \oplus \Sigma_2$ such that $\Sigma_2$ is diagonal and

$$\frac{\text{rank}(\Sigma_1)}{n} \to 0, \qquad \frac{\|w^*\|_1 \mathbb{E} \|\Sigma_2^{1/2} H\|_\infty}{\sqrt{n}} \to 0, \qquad \frac{n}{r_1(\Sigma_2)} \to 0. \tag{17}$$

**Application: Junk features.** We now consider the behavior of basis pursuit in a junk feature model similar to that of Zhou et al. (2020). Suppose that $\Sigma = \begin{bmatrix} \Sigma_s & 0 \\ 0 & \frac{\lambda_n}{\log(d)} I_d \end{bmatrix}$, where $\Sigma_s$ is a fixed matrix and $\|w^*\|_1$ is fixed. Quite naturally, we choose the covariance splitting $\Sigma_1 = \begin{bmatrix} \Sigma_s & 0 \\ 0 & 0 \end{bmatrix}$, which has constant rank so that the first sufficient condition is immediately satisfied.

By standard results on the maximum of independent Gaussian variables (e.g. Vershynin 2018), it is routine to check that

$$\frac{\mathbb{E}\,\|\Sigma_2^{1/2} H\|_\infty}{\sqrt{n}} = \Theta\left(\sqrt{\frac{\lambda_n}{n}}\right) \quad \text{and} \quad r_1(\Sigma_2) = \Theta\left(\log(d)\right). \tag{18}$$

Therefore, basis pursuit will be consistent provided that $\lambda_n = o(n)$ and $d = e^{\omega(n)}$. To the best of our knowledge, this is the first time that basis pursuit has been shown to give consistent predictions in any setting with Gaussian covariates and $\sigma > 0$. Although we show consistency, the dimension must be quite high, and the rate of convergence depends on $n/\log(d)$ and $1/\sqrt{\log(d)}$.

**Application: Isotropic features.** As in the Euclidean case, we generally do not expect basis pursuit to be consistent when $\Sigma = I_d$ and $w^* \neq 0$. However, we can expect its risk to approach the null risk $\sigma^2 + \|w^*\|^2$ if $d = e^{\omega(n)}$; we will show this using uniform convergence (without covariance splitting).

A direct application of Theorem 5 is not enough because the $\|w^*\|_1 \sqrt{\log(d)/n}$ term diverges, but we can remove the dependence on $\sqrt{\log(d)/n}$ with a better norm bound. Let $S$ be the support of $w^*$ and denote $X_S$ as the matrix formed by selecting the columns of $X$ in $S$. The key observation is that we can rewrite our model as $Y = X_{S^c} 0 + (X_S w_S^* + \xi)$, which corresponds to the case when $w^* = 0$ and the Bayes risk is $\sigma^2 + \|w^*\|_2^2$. If we interpolate using only the features in $S^c$, the minimal norm will be approximately upper bounded by $\sqrt{\sigma^2 + \|w^*\|_2^2}\,\frac{\sqrt{n}}{\mathbb{E}\,\|H\|_*}$ as long as $d - |S| = e^{\omega(n)}$, by Theorem 4. This implies the original model $\|\hat{w}_{BP}\|_1$ can also be upper bounded by the same quantity with high probability. Plugging the norm estimate in to Corollary 3 yields a risk bound of $\sigma^2 + \|w^*\|_2^2$.

**Relationship to previous works.** Both Ju et al. (2020) and Chinot et al. (2021) study the minimal $\ell_1$ norm interpolator in the isotropic setting. They consider a more realistic scaling where $\log(d)/n$ is not large and the target is not the null risk. The best bound of Ju et al. (2020), their Corollary 3, is $L(\hat{w}_{BP}) \leq \sigma^2(2 + 32\sqrt{14}\sqrt{s})^2$, where $s$ is the ground truth sparsity. Note that even when $w^* = 0$, this bound does not show consistency. Similarly, Chinot et al. (2021) establish sufficient conditions for $L(\hat{w}_{BP}) = O(\sigma^2)$, which is nontrivial but also does not show consistency for any $\sigma > 0$; see also the work of Chinot and Lerasle (2020) for a similar result in the Euclidean setting. In contrast, the constants in our result are tight enough to show $L(\hat{w}_{BP}) \to \sigma^2$ in the isotropic setting when $w^* = 0$ and in the junk feature setting when $\|w^*\|_1$ is bounded.

Like our work, the results of Chinot et al. (2021) generalize to arbitrary norms; they also consider a larger class of anti-concentrated covariate distributions than just Gaussians, as in the work of Mendelson (2014). If $\sigma = 0$ and $w^* \in \mathcal{K}$ (i.e. the model is well-specified and noiseless), their work as well as that of Mendelson (2014) can recover generalization bounds similar to our Corollary 3, but with a large leading constant.

# 7 Proof Sketches

A key ingredient in our analysis is a celebrated result from Gaussian process theory known as the Gaussian Minmax Theorem (GMT) (Gordon 1985; Thrampoulidis et al. 2015). Since the seminal work of Rudelson and Vershynin (2008), the GMT has seen numerous applications to problems in statistics, machine learning, and signal processing (e.g. Oymak and Hassibi 2010; Stojnic 2013; Oymak and Tropp 2018; Deng et al. 2021). Most relevant to us is the work of Thrampoulidis et al. (2015), which introduced the Convex Gaussian Minmax Theorem (CGMT) and developed a framework for the precise analysis of regularized linear regression. Here we apply the GMT/CGMT to study uniform convergence and the norm of the minimal norm interpolator.

**Proof sketch of Theorem 1.** For simplicity, assume here there is no covariance splitting: $\Sigma_2 = \Sigma$. By a change of variable and introducing the Lagrangian, we can rewrite the generalization gap as

$$\sup_{\substack{w \in \mathcal{K} \\ Xw=Y}} L(w) = \sigma^2 + \sup_{\substack{w \in \Sigma^{1/2}(\mathcal{K}-w^*) \\ Zw=\xi}} \|w\|_2^2$$

$$= \sigma^2 + \sup_{w \in \Sigma^{1/2}(\mathcal{K}-w^*)} \inf_\lambda \langle \lambda, Zw - \xi \rangle + \|w\|_2^2 \tag{19}$$

where $Z$ is a random matrix with i.i.d. standard normal entries. By GMT[3], we can control the upper tail of the max-min problem above (PO) by the auxiliary problem below (AO), with $G, H \sim N(0, I)$:

$$\sup_{w \in \Sigma^{1/2}(\mathcal{K}-w^*)} \inf_\lambda \|\lambda\|_2 \langle H, w \rangle + \langle \lambda, G\|w\|_2 - \xi \rangle + \|w\|_2^2 = \sup_{\substack{w \in \Sigma^{1/2}(\mathcal{K}-w^*) \\ \|G\|w\|_2-\xi\|_2 \leq \langle H,w \rangle}} \|w\|_2^2. \tag{20}$$

By standard concentration results, we can expect $\|G\|_2^2/n \approx 1$ and $\|\xi\|_2^2/n \approx \sigma^2$, so expanding the second constraint in the AO, we obtain $\|w\|_2^2 + \sigma^2 \leq |\langle H, w \rangle|^2/n$. Plugging into (19), we have essentially shown that

$$\sup_{\substack{w \in \mathcal{K} \\ \hat{L}(w)=0}} L(w) \leq \sup_{w \in \Sigma^{1/2}(\mathcal{K}-w^*)} \frac{|\langle H, w \rangle|^2}{n} \leq \frac{\left(\sup_{w \in \Sigma^{1/2}\mathcal{K}} |\langle H, w \rangle| + |\langle H, \Sigma^{1/2}w^* \rangle|\right)^2}{n}. \tag{21}$$

Applying concentration on the right hand side concludes the proof sketch. In situations where the supremum does not sharply concentrate around its mean, we can apply GMT only to the small variance directions of $\Sigma$. This requires a slightly more general version of GMT, which we prove in Appendix A. We also show the additional terms contributed by the large variance components of $X$ cancel out due to Wishart concentration. This is reflected in the $\beta$ term of our theorem statement.

**Proof sketch of Theorem 4.** Since the minimal norm problem is *convex-concave*, we can apply the CGMT, which provides a useful direction that GMT cannot. By the same argument as above

$$\inf_{Xw=Y} \|w\| - \|w^*\| \leq \inf_{Xw=\xi} \|w\| = \inf_w \sup_\lambda \|\Sigma^{-1/2}w\| + \langle \lambda, Zw - \xi \rangle$$

$$\approx \inf_{\|w\|_2^2+\sigma^2 \leq |\langle H,w \rangle|^2/n} \|\Sigma^{-1/2}w\| = \inf_{\|w\|_\Sigma^2+\sigma^2 \leq |\langle \Sigma^{1/2}H,w \rangle|^2/n} \|w\|. \tag{22}$$

To upper bound the infimum, it suffices to construct a feasible $w$. Consider $w$ of the form $\alpha v$ where $v \in \partial\|\Sigma^{1/2}H\|_*$. Plugging in the constraint, we can choose $\|w\| = \alpha = \sqrt{\sigma^2 \left(\frac{\|\Sigma^{1/2}H\|_*^2}{n} - \|v\|_\Sigma^2\right)^{-1}}$.

Rearranging the terms conclude the proof sketch when there is no covariance splitting. The general proof (in Appendix C.1) is more technical, but follows the same idea.

# 8 Discussion

In this work, we prove a generic generalization bound in terms of the Gaussian width and radius of a hypothesis class. We also provide a general high probability upper bound for the norm of the minimal norm interpolator. Combining these results, we recover the sufficient conditions from Bartlett et al. (2020) in the $\ell_2$ case, confirm the conjecture of Zhou et al. (2020) for Gaussian data, and obtain novel consistency results in the $\ell_1$ case. Our results provide concrete evidence that uniform convergence is indeed sufficient to explain interpolation learning, at least in some settings.

A future direction of our work is to extend the main results to settings with non-Gaussian features; this has been achieved in other applications of the GMT (Oymak and Tropp 2018), and indeed we expect that a version of ($\star$) likely holds for non-Gaussian data as well. Another interesting problem is to study uniform convergence of low-norm *near*-interpolators, and characterize the worst-case population error as the norm and training error both grow. This could lead to a more precise understanding of early stopping, by connecting the optimization path with the regularization path. Finally, it is unknown whether our sufficient conditions for consistency in Section 5 are necessary, and it remains a challenge to apply uniform convergence of interpolators to more complex models such as deep neural networks.

---

[3]We ignore a compactness issue here, but this is done rigorously by a truncation argument in Appendix B.1.

## Acknowledgments and Disclosure of Funding

Frederic Koehler was supported in part by E. Mossel's Vannevar Bush Faculty Fellowship ONR-N00014-20-1-2826. Research supported in part by NSF IIS award 1764032, NSF HDR TRIPODS award 1934843, and the Canada CIFAR AI Chairs program. This work was done as part of the Collaboration on the Theoretical Foundations of Deep Learning (`deepfoundations.ai`).

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
