# OpenReview forum: "Uniform Convergence of Interpolators: Gaussian Width, Norm Bounds and Benign Overfitting"
_NeurIPS.cc/2021/Conference — NeurIPS 2021 Oral_

### Official Review · Reviewer_CCVZ · 2021-07-16

**Rating:** 8
**Confidence:** 4

**Summary:**

The authors study generalization bounds for minimum norm linear interpolants. They use techniques from uniform convergence to derive a general bound that applies to minimum norm linear interpolant, that depends on both the norm of the optimal linear predictor as well as the Gaussian width and radius of the corresponding norm ball. In the case of the Euclidean norm the authors re-derive the sufficient conditions previously obtained by Bartlett et al. (2020) by deriving an upper bound on the norm of the predictor. In the case of the l_1 norm, the authors provide a set of sufficient conditions for the minimum norm interpolant to be consistent and discuss the implications in two special cases.

**Limitations And Societal Impact:**

The authors have addressed the limitations and potential impacts of their work.

**Main Review:**

This paper was a pleasure to read, and the authors provide an interesting set of results that help improve our understanding of the behavior of interpolating models. Their results also indicate that the fact that techniques from uniform convergence may indeed be useful in understanding the generalization behavior of interpolating models by skirting past recent lower bounds (e.g. Bartlett and Long).  Some questions for the authors that hopefully shall help improve this draft:

1. In ln 107 the authors say that we expect the Gaussian width to scale like sigma sqrt{n}. I dont see where the dependence on sigma comes in here if the set K is just the norm ball of radius B.

2. In equation 12 is it clear if the sufficient conditions for the l_1 minimizer to be consistent close to necessary? Is there any evidence, theoretical or otherwise to suggest this or not?

3. On ln 266 the authors say ||w^*||_1 sqrt{log(d)/n} diverges, but it is not clear to me why this is the case.

4. I would appreciate a more detailed comparison between the techniques used in this paper and the techniques employed by Ju et al. and Chinot et al. who also study the minimum ell_1 norm interpolant using uniform convergence techniques.


5. On ln 297, both equation (PO) and equation (AO) are not defined.


========== Post rebuttal comments ==========

I thank the authors for their responses. I would like to continue to vote for acceptance, and would be happy to see this paper being published at NeurIPS.


**Time Spent Reviewing:**

2

---

> ### Author Response · Authors · 2021-08-10
> **Response**
>
> Thank you for your work in reviewing our paper.
>
> > 1. ln 107 the authors say that we expect the Gaussian width to scale like sigma sqrt{n}. I dont see where the dependence on sigma comes in here if the set K is just the norm ball of radius B.
>
> For a fixed $B$, indeed the Gaussian width does not depend on $\sigma$ (or $n$). The dependence here comes up because we’re assuming $B$ is chosen to be (at least) the norm of the minimal-norm interpolator, in order for the theorem to be nontrivial; if there’s more noise to interpolate (higher $\sigma$), more norm is required to cover it. We’ll clarify this discussion.
>
> > 2. In equation 12 is it clear if the sufficient conditions for the l_1 minimizer to be consistent close to necessary? Is there any evidence, theoretical or otherwise to suggest this or not?
>
> We don’t think (near-)necessity is clear here in general, and basically are not sure. (Note that there is still a gap between known necessary and sufficient conditions even in the Euclidean case after significant prior study [3, 22, 30], although that gap is certainly smaller than we leave for the l1 conditions.) We do expect that $d =e^{\omega(n)}$ should usually be necessary for consistency in the l1 setting: to sparsely fit the noise, we need “enough chances” at features that align well with the noise vector (angular distance $o(1)$), and a covering number argument for the sphere in $\mathbb{R}^n$ gives the threshold $d = e^{\omega(n)}$.
>
>
> > 3. On ln 266 the authors say ||w^*||_1 sqrt{log(d)/n} diverges, but it is not clear to me why this is the case.
>
> Here we mean specifically when $d = e^{\omega(n)}$, so that $\log d = \omega(n)$ and $\sqrt{\log(d) / n} \to \infty$.
>
>
> > 4. I would appreciate a more detailed comparison between the techniques used in this paper and the techniques employed by Ju et al. and Chinot et al. who also study the minimum ell_1 norm interpolant using uniform convergence techniques.
>
> Comparison to Ju et al. [17]: As we discuss around line 274, Ju et al. do not obtain sharp constants in this setting and cannot show consistency for $\sigma > 0$. In terms of proof technique: their core strategy is to bound the error in terms of the 1-norm of the predictor that interpolates only noise and the incoherence of the design matrix. They show that the noise can be interpolated with a rate depending on $\sqrt{n / \log d}$ – related to our heuristic covering numbers argument. They also bound the incoherence based on the maxima of Gaussians. They do not explicitly frame their argument in terms of uniform convergence, and don’t prove any results about interpolators other than the exact minimum l1-norm predictor.
>
> Comparison to Chinot et al. (‘On the Robustness of Minimum-Norm Interpolators’, [8]): As we mentioned in the paper, the key difference in terms of results is that they prove bounds of the form $O(\sigma^2)$, so their results do not establish consistency for $\sigma > 0$, whereas our results do. In terms of techniques, they do use a uniform convergence argument somewhat like ours, combining a generalization bound with a norm bound. The main technical difference is that in our main result we prove a new generalization bound with sharp constants for Gaussian data, whereas for getting $O(\sigma^2)$ they observed that the existing generalization bounds due to Mendelson [20] / Lecue and Mendelson, for any anti-concentrated data distribution, are sufficient. (They also gave a variant of those results to handle other assumptions on the noise.) Similarly, using CGMT we obtain norm bounds with sharp constants, which they did not (need to) obtain.
>
>
> > 5. On ln 297, both equation (PO) and equation (AO) are not defined.
>
> Sorry for the confusion here; PO refers to the max-min optimization problem inside (16), and AO to (17). We didn’t mean “(PO)” to be an equation reference, but rather just establishing a “nickname” for the Primary Optimization and the Auxiliary Optimization, as is often done in treatments of the CGMT. We’ll rephrase to clarify this.

---

### Official Review · Reviewer_Gkhk · 2021-07-19

**Rating:** 8
**Confidence:** 4

**Summary:**

This paper uses (a small modification of) the CGMT for a uniform bound and the GMT for bounding the norm of the min-norm linear interpolators for general norms. Using this technique they recover generalization bounds for l2 and l1 and consistency conditions such as in Tsigler, Bartlett et al. or Zhou et al. In contrast to these works their bound does not only apply to the exact min-norm interpolator but for all linear interpolators that have a norm as small as an high probability upper bound of the min-norm interpolator. By doing this they provide a counterpoint to some recent works that question uniform convergence to explain the generalization capabilities of overparameterized interpolating models.

**Ethical Concerns:**

-

**Limitations And Societal Impact:**

-

**Main Review:**

Strengths:
- the authors provide a novel and (to the best of my knowledge) innovative and elegant way to use uniform bounds for low-norm linear interpolators for regression (proven separately before for the min-l1 and min-l2) by appealing to the (C)GMT.
- the manuscript is very clearly and nicely written
- the proof sketch contains the key ideas of the proof so it’s very informative

Constructive comments:
- The proof technique is the key novelty and one certainly wonders about the impact of the exercise of unifying proof framework for l1 and l2 generalized to norms in general -> is there any gain going forward for non-Gaussian distributions, neural networks etc. - the impact of the new technique is not quite clear to me since the (C)GMT crucially rely on Gaussianity. On the other hand, the l1-discussion about w* = 0 - relatively speaking, it’s not that big of a result IMO. Hence in my view, given the limited space, it would be much more enlightening to shorten that section and instead discuss the impact of the new proof technique. Also a discussion about more insights from the proof would be very helpful / useful - what is the intuition behind why consistency for min-l1 can be achieved for spiked covariances with d = e^{omega(n)}?
- The proof would benefit from significant modularization. Navigating through the proofs is unnecessarily cumbersome since there is a lot of nesting, lemmas are often proved along the way

On  another note:
Regarding the story that makes a case for/in defense of uniform convergence min-norm-ball: I guess somewhat concurrently, Chinot et al. ‘21 (AdaBoost ...) have also used uniform convergence techniques for min-l1-norm classifiers although they study a different setting …


**Time Spent Reviewing:**

10

---

> ### Author Response · Authors · 2021-08-10
> **Response**
>
> Thank you for your work in reviewing our paper. Your questions about non-Gaussian distributions and impact of the proof technique are, we think, exactly the right questions to ask for follow-up work. In large part, we view our new results as a proof of concept showing that analysis with “the right type” of uniform convergence is capable of analyzing interesting problems – problems that, even after Zhou et al. [38] argued for this type of convergence in the wake of Nagarajan and Kolter [21], may still have seemed impossible after cursory readings of Bartlett and Long [2] or Yang et al. [36]. The proof technique used here does indeed rely heavily on having Gaussian data and squared loss; it is not yet clear if the approach can be significantly generalized beyond that setting. Investigating uniform convergence-based interpolation learning in a non-Gaussian setting is indeed an important question we are working on, and hope others are as well – perhaps the most immediate question being whether the speculative bound (*) holds with non-Gaussian data. We view our main message, though, as continuing the discussion on the possibilities of uniform convergence, and advocating for the notion of uniform convergence of interpolators. We are hopeful that, although there will surely need to be innovations in techniques to get there, this type of analysis will eventually be able to explain practical deep learning settings.
>
> Spiked covariances: intuitively, since the l1 norm encourages sparsity, we need to fit the noise efficiently with a sparse vector, and so we need enough random chances at a feature that will by chance mostly align with the noise. Based on the covering number of the sphere in $\mathbb{R}^n$, once we pass the threshold $d = e^{\omega(n)}$ there is likely to be a feature pointing in almost the same direction as the noise vector (i.e., the angular distance is $o(1)$).
>
> > The proof would benefit from significant modularization. Navigating through the proofs is unnecessarily cumbersome since there is a lot of nesting, lemmas are often proved along the way
>
> We totally agree, and in fact have already (after submission) re-organized the appendix to flow more smoothly. We think it’s much easier to read this way; sorry that you had to struggle through the current complicated version!
>
> > On another note: Regarding the story that makes a case for/in defense of uniform convergence min-norm-ball: I guess somewhat concurrently, Chinot et al. ‘21 (AdaBoost … [https://arxiv.org/abs/2105.02083]) have also used uniform convergence techniques for min-l1-norm classifiers although they study a different setting …
>
> Thanks for introducing us to this (concurrent) work. It seems that they establish consistency in the agnostic classification setting with o(1) corrupted labels; the linear regression analogue would seem to be the previous work of Chinot et al. [8], giving $O(\sigma^2)$ guarantees which imply consistency when $\sigma = o(1)$. In contrast, the key contribution of our work is to establish consistency via uniform convergence even when $\sigma > 0$, where a $O(\sigma^2)$ guarantee is not sufficient. It is an interesting direction for future work to consider analogous results for classification. In terms of the proof technique, at first glance their methods appear rather different from ours, but we’ll take a little time to absorb and add a discussion in revision.

---

### Official Review · Reviewer_GGnj · 2021-07-20

**Rating:** 7
**Confidence:** 3

**Summary:**

This article deals with excess risk guarantees, under Gaussian noise, of linear regression with interpolators: estimators with zero training error. In particular, it highlights the role played by the norm of an interpolator in the generalization bounds. The proposed analysis covers a wide range of problems since it is valid for a large class of norms. In particular, the analysis covers the case of  $\ell_1$ norm interpolator and the  $\ell_2$ norm interpolator. These upper bounds were used to prove the consistency of the respective interpolators.


**Limitations And Societal Impact:**

No.

**Main Review:**

Although heavily technical, the article is well written and the proven results were widely discussed and compared to the existing results in the literature. However, some aspects of this work need to be clarified:
- the theoretical results proven in this article are relevant and single out from existing work in the regime  $n$ goes to infinity. Yet, in practice, the dimension d  is finite. Taking both n and d to infinity makes sense in the context of kernel-based interpolation: the effective dimension increases with n and goes to infinity as the regularisation parameter goes to $0$. But, it seems to be unnatural to take d to infinity in the context of linear regression. How can we conciliate the two seemingly contradictory regimes (n goes to infinity and d is finite) in the context of linear regression?
- It would be interesting to give the intuition behind the conditions for consistency in equation (7), especially for the two quantities $\sqrt{\frac{Tr \Sigma_2}{n}}$ and $\frac{n}{R(\Sigma_2)}$. (Or at least the order of magnitude of these quantities when $\Sigma$ is a sequence of decreasing exponentially or polynomially to 0.)
-The same goes with the conditions of Theorem 2: intuitively, what does it mean to have $R(\Sigma_2) \geq n \log(2/\delta) ...$?

Minor comments:
- Line 122: I can’t understand why the rate is only $\mathcal{O}(N^{-1/4})$ since $1+\beta^{,}$ converges to 1.
- Equation (7) is misleading: we may think that the three terms are equals...


**Time Spent Reviewing:**

10

---

> ### Author Response · Authors · 2021-08-10
> **Response**
>
> Thank you for your work in reviewing our paper.
>
> In linear regression, interpolation requires that $d \ge n$, and so to study interpolation as $n \to \infty$ also requires that $d \to \infty$, as has been done by all prior work of interpolation learning in linear regression [e.g. 3, 8, 10, 11, 16, 17, 22, 30, 36, 38]. This roughly corresponds to studying “what happens when both dimension and sample size are very large,” with some relationship depending on the particular mode of study. Our results, unlike some previous work in this area that study only asymptotics, actually hold for finitely many samples $n$ in finite dimension $d$, so that it is possible with our bounds to study situations with any relationship between $n$ and $d$ – bearing in mind that when $d \le n$ there simply are no interpolators and our object of study becomes trivial.
>
> The rank conditions in the l2 case (7) and the closely related Theorem 2 are due to Bartlett et al. [3], who give examples of problem sequences where these conditions are satisfied or not, such as when the eigenvalues of $\Sigma$ follow various rates of decay. Our purpose in establishing the conditions here is to demonstrate that the “uniform convergence of interpolators” approach is capable of obtaining the same sufficient conditions as was previously obtained by other methods; it is not a new condition about consistency of interpolators. The general and l1 counterparts are novel to us; in some sense the purpose of Section 6 is to establish examples for when these conditions hold.
>
> Line 122: In e.g. the setting of Zhou et al. [38], or indeed the result of Theorem 2 to come, picking $B = \lVert \hat{w} \rVert_2^2$ gives that $B^2 \operatorname{Tr}(\Sigma) / n \approx \sigma^2$, the Bayes error which we are trying to obtain. Then the RHS of (3) becomes roughly $(1 + \sqrt[4]{\log(32/\delta)/n} ) \sigma^2$, and so it converges at a rate of $n^{-1/4}$ to the desired value $\sigma^2$. We’ll clarify this in the paper.
>
> (7), and also (12) and (15): We agree that they’re currently written in a confusing way; we’ll rephrase as saying that each term $\to 0$.

---

> > ### Comment · Reviewer_GGnj · 2021-08-30
> > **Score update**
> >
> > Thank you for the explanation. I updated my score to 7.

---

### Decision · Program_Chairs · 2021-09-27

**Decision:**

Accept (Oral)

**Comment:**

The paper studies uniform convergence for linear predictors in the high-dimensional setting with gaussian assumptions on the data. This setting has been adopted by the learning theory community to tractably study generalization phenomena arising in deep learning. In this setting, the paper gives new bounds on the generalization error of interpolating solutions, where the bound depends on the gaussian width of the function class and the norm of the predictor. This recovers some existing bounds derived in special cases, and adds a counterpoint to recent work suggesting uniform convergence cannot explain generalization in high dimensional settings.

All reviewers agreed that the results are novel and interesting, the paper is clearly written, and the proofs are easy to follow and illuminating. As such, we believe this will make a strong contribution to the program.